# Diversifying Forest Landscape Management—A Case Study of a Shift from Native Forest Logging to Plantations in Australian Wet Forests

David Lindenmayer *[ORCID] and Chris Taylor

Fenner School of Environment and Society, The Australian National University, Canberra, ACT 2601, Australia; christopher.taylor@anu.edu.au
* Correspondence: david.lindenmayer@anu.edu.au

**Abstract:** Natural forests have many ecological, economic and other values, and sustaining them is a challenge for policy makers and forest managers. Conventional approaches to forest management such as those based on maximum sustained yield principles disregard fundamental tenets of ecological sustainability and often fail. Here we describe the failure of a highly regulated approach to forest management focused on intensive wood production in the mountain ash forests of Victoria, Australia. Poor past management led to overcutting with timber yields too high to be sustainable and failing to account for uncertainties. Ongoing logging will have negative impacts on biodiversity and water production, alter fire regimes, and generate economic losses. This means there are few options to diversify forest management. The only ecologically and economically viable option is to cease logging mountain ash forests altogether and transition wood production to plantations located elsewhere in the state of Victoria. We outline general lessons for diversifying land management from our case study.

**Keywords:** sustainable forest management; forest history; pattern and process; fire regimes; biodiversity; ecosystem values

## 1. Introduction

The world's forests play critical roles in water cycles, carbon storage, wood production, and biodiversity conservation [1–3]. How forests are managed can have profound impacts on these roles, particularly where uses such as wood production conflict with other goals such as the protection of biodiversity [4], the maintenance of carbon stocks [5,6] or the supply of water for human consumption [7]. Long-term maintenance of the range of values of natural forests is a key part of ecologically sustainable forest management [8,9], where ecologically sustainable forest management can be broadly defined as:

> *Forest management that perpetuates ecosystem integrity while continuing to provide wood and nonwood values. In this context, ecosystem integrity can be considered to be the maintenance of forest structure, species composition, and the rate of ecological processes and functions within the bounds of normal disturbance regimes.*

Whilst ecologically sustainable forest management is the goal of management agencies in many parts of the world [9], it can be difficult to achieve for a wide range of reasons, particularly the array of ecological, economic, silvicultural, social and other factors that need to be considered [8,10,11]. Ecologically sustainable forest management is particularly difficult to achieve when wood production is based on principles of maximum sustained yield or the highly regulated forest concept (sensu [12]).

The 'regulated' forest or 'normal' forest concept has long been a focus of conventional forest management [12,13], and it has been a long-standing legacy of so-called "colonial forestry" [14]. The uniformity of a normal forest was intended to facilitate management



and extraction, and it became a powerful aesthetic for a well-managed forest [15]. In 1849, Faustmann (translated in [16], cited by [17]) used the concept of a normal forest to calculate a logging rotation period that would maximise economic benefits. In the simple case of a single commercial tree species, uniform site conditions and a single silvicultural system, the harvesting regimes can be expressed as after [13]:

$$\text{Total area/rotation age = area in each age class}$$

The regulated forest was an abstraction and an attempt to rationalize nature and make it knowable, calculable and visible [18]. Strict application of this simple equation would result in roughly equal areas of each age class in a given ecosystem. This management strategy has at its core, the aim of maximizing economic benefits as well as the output of forest products that can theoretically be sustained over time (e.g., [19]). The ultimate objective is the perpetual, even flow of wood products for a forest industry [2].

A major problem with the regulated forest concept in forestry is that it is focuses on resource exploitation [11] and ignores the inherent social and environmental complexities of forests [14,15]. It also ignores uncertainty that may arise from measurement error, natural variation that affects the distribution and abundance of the resource (e.g., the impacts of disturbance on wood stocks, such as fire) and a lack of understanding of the ecology of the species. Failure to account for stochasticity or other factors means that estimates of sustained yield do not have sufficient 'ecological margins' to accommodate such impacts on the stock available for logging. A further problem with the regulated forest approach is that it focuses almost exclusively on wood production, and other values, including key ecosystem services, are given limited consideration, thereby contravening the overarching objectives of ecologically sustainable forest management [8,11,20].

The regulated forest was often placed under the management of centralized forest government bureaucracies [14]. In some cases, such as in Australia, original centralization of forestry was undertaken to address other issues of unsustainable practices such as widespread forest clearing for agriculture and grazing [21]. The centralization of forestry was a central tenet of empire forestry [22]. As colonialism expanded across Africa, Asia, Australia and the Americas throughout the 18th and 19th centuries, centralized colonial bureaucracies assumed control over vast forest areas, often excluding local people from their lands and suppressing traditional forest institutions [23]. The tenure of forests changed from local communities to distant state agencies. Government at a distance became the model of management [24]. Specific types of knowledge and techniques were imposed on forests to make them visible, calculable and therefore exploited by distant agencies [14]. The successors of these colonial regimes remain in place to this day in many parts of the world [23].

A landscape-scale and decentralized approach to forest management is sometimes suggested as a way to balance different (and often competing) forest values [11,14]. Under such an approach, a diversity of management strategies is employed in which different values are prioritized in different parts of landscapes across different communities, theoretically enabling a much wider array of values to be maintained across the broader forest estate [2,25]. For example, biodiversity and key ecosystem processes may be maintained in the face of ongoing wood production [11]. The engagement of civil society in such a context can provide for a broader basis in knowledge about forest ecosystems, which can assist in more adaptive approaches to forest management [2]. However, historic forms of centralized governance have often excluded local stakeholders, therefore leading to increasing conflict around forest management decisions [26].

In this paper, we discuss how a legacy of past forest management practices, including adoption of the normal or regulated forest model for maximum sustainable timber yields, can preclude attempts to diversify future forest management. We support our discussion with a case study on the mountain ash (*Eucalyptus regnans*) forests of the Central Highlands of Victoria (southeastern Australia). In this case, past adherence to intensive wood production of the regulated forest at the expense of other values, coupled with timber stock

losses following widespread recurrent wildfires, means there are now very few options for diversifying forest management strategies within that ecosystem. Thus, the ecosystem is in urgent need of protection and restoration to maintain biodiversity, recover highly depleted levels of old growth cover, restore natural fire regimes and maintain the security of water supplies for human consumption.

New policies are urgently required to reform forest governance and rapidly transition industrial wood production away from native forests dominated by mountain ash, toward well managed plantations and non-threatened forest ecosystems. This will spare the mountain ash ecosystem from industrial wood production with decentralized conservation strategies forming an important component of forest management. Wood production would then be transitioned into other areas that include sustainably managed plantations, and agroforestry. Our detailed case study of mountain ash forests provides some important general lessons about the pitfalls of a focus on an intensively regulated and centralized forest-based management on maximum yield principles and its implications for diversifying forest landscape management. We discuss these lessons in the concluding parts of the paper.

## 2. Study Area and Study System

The mountain ash forests of the Central Highlands of Victoria, southeastern Australia (Figure 1) cover ~140,000 ha and have been the focus of detailed ecological, silvicultural, economic and social science studies for more than four decades [27–30]. Mountain ash forests are spectacular and support the tallest flowering trees on Earth (approaching 100 m in height) [27]. These forests produce, capture and filter most of the water for the more than five million inhabitants of Melbourne [7,31], the second largest city in Australia. Mountain ash forests are important for biodiversity, including a range of threatened, endangered and critically endangered species [28,32]. These forests are important for Aboriginal people, such as the GunaiKurnai, Taungurung and Wurundjeri peoples [33–35]. Old growth mountain ash forests store large amounts of carbon and are among the most carbon-dense forests in the world [36]. The ash-type forests of the Central Highlands of Victoria (which include mountain ash forests) currently support approximately 65% of all native forest logging in the state of Victoria [30], with the majority of timber going into the pulpwood and woodchip stream [29]. Finally, the Central Highlands region in which mountain ash forests are located is also important for tourism [29].

The land tenure of mountain ash forests in the Central Highlands of Victoria consists largely of state forests (~92,000 ha) and national parks (~38,000 ha) (where logging is not permitted) [37]. Land management resides with Parks Victoria for national parks and the Department of Environment, Land, Water and Planning (DELWP) for state forests [38]. Approximately 60,000 ha of mountain ash forest from the state forests has been allocated to the Victorian government's logging business, VicForests, for the purposes of logging for pulp logs and timber [37,39]. DELWP is the land manager and service provider for state forests, where it supports the government in setting and determining policy [40].

### 2.1. Natural and Human Disturbance Regimes in Mountain Ash Forests

Wildfires and logging are the primary forms of natural and human disturbance, respectively, in mountain ash forests. Mountain ash trees are obligate-seeders and depend on specific fire regimes to regenerate and maintain their functional integrity. Mountain ash trees are typically killed by high severity wildfire but regenerate rapidly after fire from canopy-stored seed, often as even-aged cohorts of trees [41]. Understory elements such as tree ferns are long-lived and often survive successive fires [42]. Severe wildfires are typically stand-replacing events with the nominal natural inter-fire interval being 75–150 years [43], although fire frequency can range from 30 to 300+ years depending on the location in the landscape. Inter-fire intervals appear to be shortening as a result of climate change [44]. Frequent, high severity wildfires (<20–30 intervals) can preclude stands from reaching sexual maturity and developing viable seed stores [45], resulting in

regeneration failure and eventual ecological collapse [46,47]. This can result in mountain ash forests being replaced by Acacia spp. woodland [48]—sometimes termed the "interval squeeze" problem [30,49]. Notably, there is no evidence of extensive fire management or cultural burning by Aboriginal people in mountain ash forests across the Central Highlands. However, there is evidence of past cultural burning in other surrounding forest types, particularly drier mixed species forests and woodlands [50].

Logging is the primary form of human disturbance in mountain ash forests. Mountain ash forests have been logged for more than 150 years [51]. Clear-cutting followed the regulated forest concept, and it has been the conventional silivicultural system employed for the past 50 years [52]. The Victorian government allows logging to occur under timber release plans, which specify the location and gross size of "cutblocks" or harvest units as well as where logging exclusion areas occur (e.g., [53]).

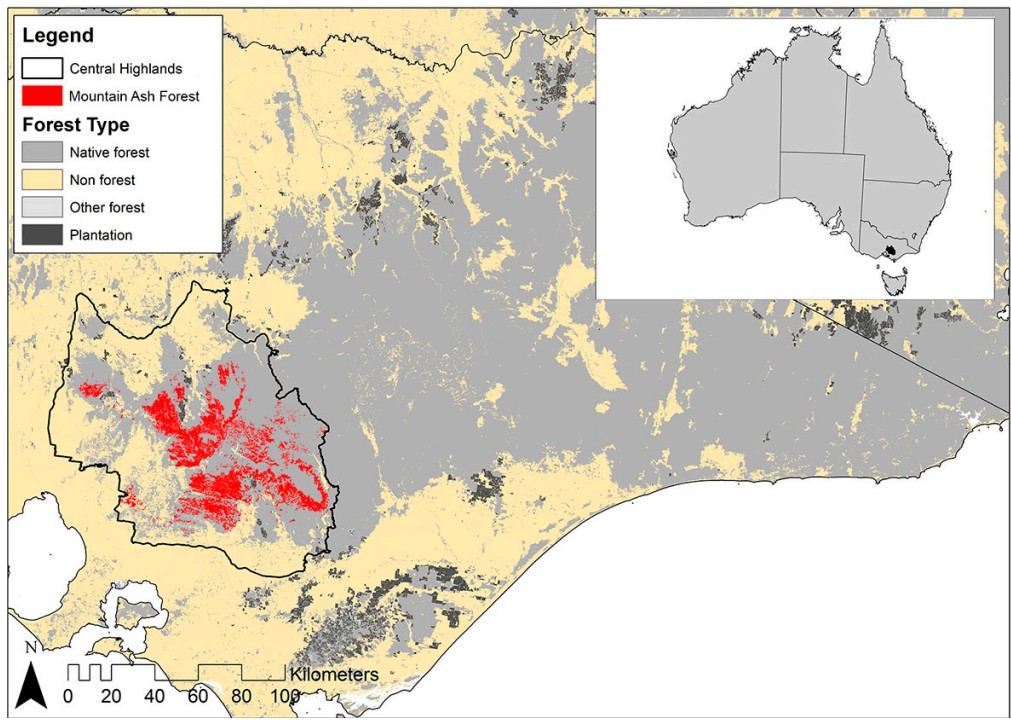

**Figure 1.** The location of the Central Highlands of Victoria, southeastern Australia.

## 2.2. 'Normalising' Mountain Ash Forests

Mountain ash forests have long been managed under a regime of widespread, industrial clear-fell logging. Indeed, in the 1920s the ash-type forests in the Central Highlands of Victoria supported more than 120 sawmills [51] (now there are five). Widespread clear-cutting commenced in the 1970s and was applied in several forest types in Victoria, including mountain ash forest [54]. The justification for its application in mountain ash forests was that wildfires in these forests are also stand-replacing events that produce even-aged stands [55]. Clear-cutting was initially developed in Victoria as a means of regrowing alpine ash (Eucalyptus delegatensis) trees following harvesting [56,57]. Clear-cutting was viewed as an efficient method of resource extraction that minimized costs and increased yields [58,59]. The clear-cutting approach in mountain ash forests is relatively simple and entails: (1) All merchantable trees being removed offsite for subsequent processing as sawn timber or for pulpwood; (2) Logging slash being left on the forest floor for one or more years to dry; (3) A high-intensity regeneration burn being applied to consume the logging slash and create a bed of ashes in which mountain ash seed is aerially dropped on the cutblock to instigate stand regeneration [59]. The typical size for cutblocks ranges from 15 to 40 ha; cutblocks can be aggregated up to 120 ha over a five-year period [60]. Clear-cutting resets stand age to zero with the specified rotation age until the next logging operation

being ~80 years. Ecological maturity in mountain ash forests is at least 170+ years [61] (and up to 500 years [62]).

Regulating mountain ash forests is aligned in principle with the regulated normal forest concept developed throughout Europe and exported around the world [14,15]. The approach was adopted in Victoria, where the former Forests Commission referred to large areas of mature and overmature forest which needed to be replaced by young healthy stands (cited in [34]). Forests were mapped in compartments and blocks based around the available area, productive area and the net harvestable area with wood yields then modelled and yield estimates calculated [63]. In such modelling, commercially valued trees such as mountain ash trees were visible, whereas noncommercial species, such as Acacia spp. and other understory forest components were invisible [64]. Long-term wood supply commitments were made based on this modelling, the most significant of these being the Forests (Wood Pulp Agreement) Act 1996, where fixed pulp log volumes from ash-type forests of the Central Highlands of Victoria were guaranteed by the state to a private company for 34 years (Forests (Wood Pulp Agreement) Act 1996).

### 2.3. Failure of the Regulated Forest to Produce Certainty in Sawlog Supply

The Victorian government's vision of regulating the mountain ash forest has not produced a forest capable of sustaining a yield of wood products in perpetuity. Significantly, there has been a collapse of the capacity of the forest to provide logs to industry. A major review into sawlog supply showed that previous modelling greatly overestimated the timber volume, with the implications for long-term logging capacity at the estimated rates of extraction for the Central Highlands of Victoria being ranked as weak to inadequate [63]. Even with subsequent sawlog yield reductions following that review, the legacy of historic overcutting remains in the forest, which is now interacting with a significant increase in wildfire frequency and extent [65,66]. Despite major wildfires, such as those in 2009 in which extensive areas of mountain ash forests were burned at high severity, there was limited appetite by the Government of Victoria to reduce the level of cut in mountain ash forests [67]. In its calculations of sustained yields of timber, VicForests failed to account for the inevitable losses in timber yields that would arise from wildfires [67]. As a consequence of historic overcutting and subsequent wildfires, VicForests was forced to reduce its sawlog and pulp log supply commitments [64]. VicForests sought to attribute this reduction to new requirements requiring logging to exclude areas where the critically endangered Leadbeater's possum was detected [68]. However, only 2848 ha of mountain ash and alpine ash forest previously allocated and available to VicForests was excluded from logging as a result of Leadbeater's possum detections. This equated to only 1.8% of the total area of ash forest allocated to VicForests [69,70]. Despite this, Leadbeater's possum detections highlight the risks of the regulated forest approach in remaining small areas of unlogged and unburned forest where logging has conflicted with areas of high conservation value. Proposed logging is scheduled across areas of highest priority for 70 threatened and forest-dependent species [32]. In an increasingly disturbed forest estate, remaining least disturbed areas are becoming critically important for conservation [37].

### 2.4. Signs of Ecological Collapse

Beyond the effects of regulated forest policy on the timber supply, clearfell logging has several significant negative environmental impacts in mountain ash forests. These include: (1) Eroding biodiversity such as populations of arboreal marsupials, birds and resprouting native tree ferns, shrubs and trees [71–73]; (2) Generating large amounts of carbon emissions, including smoke pollution during the burning of logging slash which greatly reduces the air-quality of surrounding communities [74]; (3) Depleting key soil nutrients and altering soil structure [75]; (4) Altering the soil microbiome such as by reducing the diversity of critically important fungal symbionts [76]; (5) Reducing water yields from watersheds [7]; (6) Fragmenting patterns of forest cover [37]; (7) Increasing levels of forest fire severity [65,77]. Some of these effects are at the stand level such

as impacts on soils [75]; others such as losses in biodiversity and altered patterns of flammability are primarily landscape level effects [30,72]. Yet other impacts are at an ecosystem level (such as logging-related elevated risks of ecosystem collapse [47]).

The ecological integrity of mountain ash forests is now extensively eroded because of widespread logging and fire. Around 70% of mountain ash forest in the Central Highlands of Victoria is either severely disturbed or is within 200 m of a severely disturbed area [37]. The amount of old growth mountain ash forest (>120 years old) has been reduced dramatically [78] relative to what it was historically (estimated to be ~30–60% of the extent of this ecosystem) [79]. Only 1.16% of the mountain ash ecosystem is now old growth. The predominance (~99%) of young, highly flammable regrowth stands (<83 years old), means there is a high risk of high severity wildfire recurring in mountain ash landscapes [30]. Logging and wildfires can produce interactive and cumulative effects in mountain ash forests. Empirical studies by [65] after the 2009 wildfires in the Central Highlands of Victoria showed that logged and regenerated forests burned at significantly higher severity than unlogged forest (Figure 2). Importantly, strategies such as thinning generally do not reduce the risk of high-severity wildfire in mountain ash forests and sometimes can exacerbate these risks [77,80].

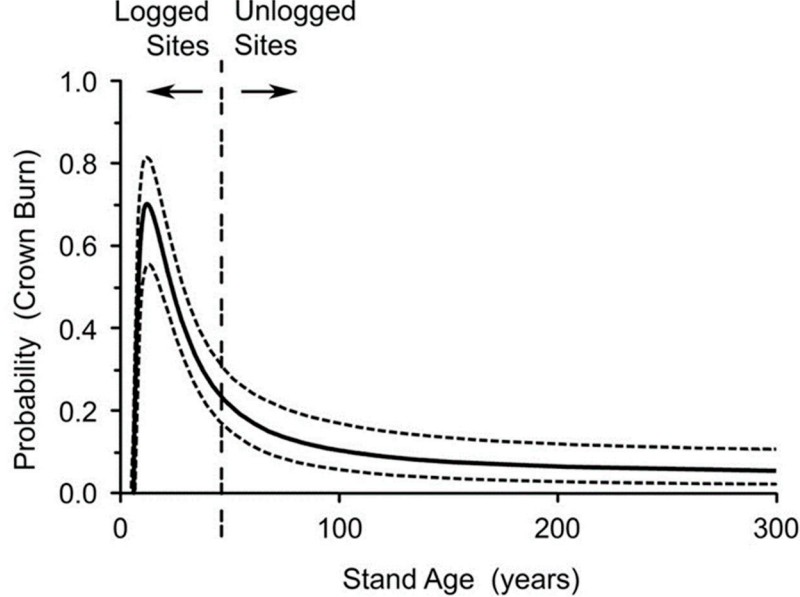

**Figure 2.** Nonlinear relationship between stand age and the probability of canopy fire or crown burn. The probability of crown burn peaks at ~40 years before declining as stands approach 80–100 years old. Redrawn from Taylor, McCarthy and Lindenmayer [65]. The solid line corresponds to mean response, and the dashed line represents the lower and upper bounds of the 95% confidence interval.

Recurrent fire means there is a high chance that the forest will reburn before trees have a chance of growing to ecological maturity of 170 years [44], a phenomenon that has been termed a "landscape trap" [30,48]. Indeed, wildfire is predicted to be so frequent that there is a high probability that trees will not reach an age (~80 years) where they will be suitable for the production of sawn timber [44]. This, in turn, will create considerable uncertainty in resource availability for an ongoing native forest timber industry in mountain ash forests [67]. Resource limitations are already leading to the weakening of forest laws and codes of practice allowing timber to now be cut from places such as very steep slopes where logging was previously banned under former (but now relaxed) codes of forest practice [81].

Logging continues to contribute significantly to the fragmentation of the mountain ash forest estate [37]. For example, the average distance from uncut areas to a disturbance boundary (a road or logged forest) in wood production mountain ash forests is just 71 m [32].

There are other consequences of the highly disturbed landscapes which characterize the mountain ash ecosystem. There have been marked declines in bird and mammal biota [47], with levels of site occupancy among some species of conservation concern such as the southern greater glider (*Petauroides volans*) having declined by ~80% in the past 20 years [72]. Moreover, logging is continuing to occur in forests of high conservation value for Victoria's threatened forest-dependent species [32]. Logging in water catchments is also having impacts on water yields, with its effects outweighing those caused by some projected climate change scenarios [7], thereby potentially compromising water security for the city of Melbourne.

Beyond the direct impacts of native forest logging in mountain ash forests on fire regimes, biodiversity and water security, there are other signs of major problems in the timber industry in Victoria. For example, there have been marked declines in employment over the past decade [82] and the Victorian government's logging company has suffered significant financial losses in most years since 2004 [83,84]. Independent economic analyses, including by the Victorian Parliamentary Budget Office, have indicated that the state of Victoria would be better off financially by between AUD $110 m [29] and AUD $190 m per year if it did not log native forests [85]. Moreover, formal environmental and economic accounting analysis has revealed marked disparities between the relative values of different natural assets in mountain ash forests [29]. For example, the value-added value to regional GDP from water is 25.5 times that of timber and woodchips from native forest logging (see Figure 2). The value of tourism is 20 times that of the native forest logging sector. Notably, the value-added value of the plantation sector in the Central Highlands region (where the environmental and economic accounting analysis was focused) was three times that of the native forest logging sector (Figure 3, [29]). Data on employment in various industries are also insightful (see [29]). Direct employment in the plantation sector in Victoria in 2012 was three times that in the native forest sector (3300 vs. 1100). Moreover, four in every five workers involved in managing, harvesting and hauling native forest timber are nonpermanent (contractor) employees. Levels of employment in the native forest sector are declining sharply [82]. As a comparator, there were 3500 tourism jobs in the Central Highlands region in 2013–2014 with the number of jobs increasing at ~5% annually. Finally, social surveys indicate that the majority of Victorians do not want native logging to continue in native forests, even in rural communities living adjacent to native forests used for wood production [86].

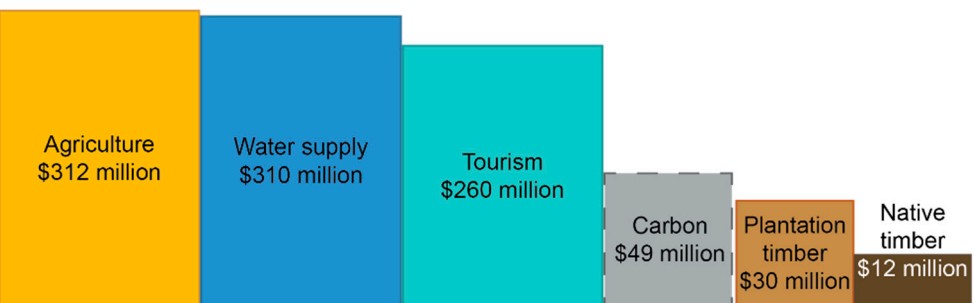

Economic contribution (Industry Value Added) to the Victorian economy from key economic activities in Victoria's Central Highlands (2013-2014). The economic contribution of agriculture, water supply, tourism, carbon and plantation timber all substantially outweigh the value of native timber production. Carbon sequestration value is an estimate of the potential value due to their being no market in Australia in 2013-2014.

**Figure 3.** Outcomes of ecosystem accounting for the key natural asset-based industries in the Central Highlands of Victoria. (Drawn from data and analyses in Keith, Vardon, Stein, Stein and Lindenmayer [29]).

## 3. Discussion: Diversifying and Decentralizing Landscape Management

A key outcome from the case study in mountain ash forests is that there has been a legacy of clinging to inappropriate and ultimately damaging historical policies such as the regulated forest concept implemented by centralized bureaucracies. The mountain ash forests of the Central Highlands are now at significant risk of ecological collapse [87], as is the timber industry that is dependent on those forests [67]. Indeed, the mountain ash ecosystem has been formally classified as critically endangered under the IUCN red listed ecosystem process because of such risks of collapse [46]. The mountain ash forest estate is so extensively altered that options to diversify landscape management are limited. Ongoing logging, irrespective of the silvicultural system used, at the current rate or even a much reduced rate, will now rapidly exhaust limited remaining timber supplies in mountain ash forests [29], add further to landscapes prone to high severity fire [65], drive down biodiversity [72] and increase levels of forest fragmentation [37].

Adherence to clearly destructive ongoing policies will mean that logging operations will often have to target increasingly marginal areas such as forest on steep slopes [88]. Moreover, areas currently proposed for logging by VicForests under approved timber release plans overlap substantially with forests of high conservation value and will therefore have major negative impacts on threatened forest-dependent biodiversity [32]. The future prognosis for the timber industry is for yet further economic losses and ongoing declines in employment. Notably, the government of Victoria has made the decision to exit the native forest logging industry not only in mountain ash forests but also in native forests across the entire state by 2030 (https://www.vic.gov.au/timber-harvesting, accessed on 28 February 2022).

Logging needs to be removed as an ecosystem stressor in mountain ash forests as part of concerted forest restoration efforts. For example, strengthened, long-term protection is needed to greatly expand the spatial extent of the old growth estate in mountain ash forests and thereby reduce forest flammability and recover key elements of biodiversity that are strongly associated with old growth forests such as the southern greater glider and the yellow-bellied glider (Petaurus australis). A long-term objective should be to restore old growth forest cover to levels that historically characterized the ecosystem—between 30 and 60% of the estate or 30–60 times more than it is currently. However, given the current risk of high severity fire across an environment that is overwhelmingly dominated by young forest (with some areas therefore very likely be burned in the coming decades), far more than 30–60% of the mountain ash ecosystem will need to be protected to reach historical targets.

### 3.1. Transition to Decentralized Forest Management

Centralized forest management regimes have often governed from a distance [18] and have lacked accountability to local communities [14]. In contrast, decentralization involves the deconcentration of administrative competencies and/or the transfer of political authority from the central state to subnational or regional administrations [23]. Decentralization has been shown, in some cases, to bring politics closer to the people, to increase policy effectiveness and to enhance democratic checks and balances at regional and community levels [23]. For the mountain ash forests, decentralization of administration provides an opportunity for Aboriginal peoples to have determination over the way their forests are managed. The mountain ash forests span several traditional countries, and each Aboriginal nation would have differing determinations for forest on their respective lands. A significant area of Australia is subject to Aboriginal management under native title [89], so these concepts are not new. Recognizing forest and land management by Aboriginal people is seen as an essential step to decolonizing conservation [90]. However, we recognize that local governance would also need to be well aligned to key governance characteristics associated with greater sustainability in resource use (e.g., greater community participation in decision making). This would ensure that there is organizational capacity and agility to adapt to key ecological, economic and social changes [91].

### 3.2. Removing Logging from Mountain Ash Forests in Victoria

Factors such as economics, biodiversity loss, fire risk, water security, carbon storage and a lack of social license all point to the need for change in the management of mainland Australian mountain ash forests. Halting logging in mountain ash forests will require finding an alternative source of wood to support forest industries in Victoria. The obvious place to find this timber is from Victoria's existing plantation sector, located primarily in other parts of the state. Other studies have shown that well managed plantations can be important for the sustainable substitution of wood production that provides offsets for enhanced conservation elsewhere [92–94]. Jurisdictions that have made a rapid transition from native forest logging to a plantation-only forest industry include New Zealand (exited native forest logging nationwide in 2002) and, recently, Western Australia which will end native forest logging at the end of December 2023 (https://www.mediastatements.wa.gov.au/Pages/McGowan/2021/09/McGowan-Governments-historic-move-to-protect-native-forests.aspx, accessed on 28 February 2022) over a transition period of 2–3 years.

### 3.3. The Availability of Plantations for Wood Supply

It has long been understood there are substantial areas of southeastern Australia suitable for plantation establishment, e.g., [95]. Data from the Australian Bureau of Agricultural and Resource Economics and Sciences (ABARES) Plantation and Log Supply inventory report show Victoria has the largest total area of forest plantations in Australia, with 385,900 hectares of commercial hardwood and softwood plantations in 2020 [96]. The plantation sector already dominates the forest industry in Victoria, in terms of volumes of sawn timber, eucalypt pulp logs, employment and economic value and returns [29,96–98] (Figures 4 and 5). In 2011, hardwood pulp log production overtook pulp log production from native forests and peaked at 3.9 million m$^3$ in 2017 and 2019 (Figure 4). In 2020, 85% of all sawn timber in Victoria was sourced from plantation estates [96]. The dominance of the plantation-based softwood sawn timber sector is evident in the Australian construction industry, where hardwood use in construction has decreased and softwood use has increased [99] (Figure 6).

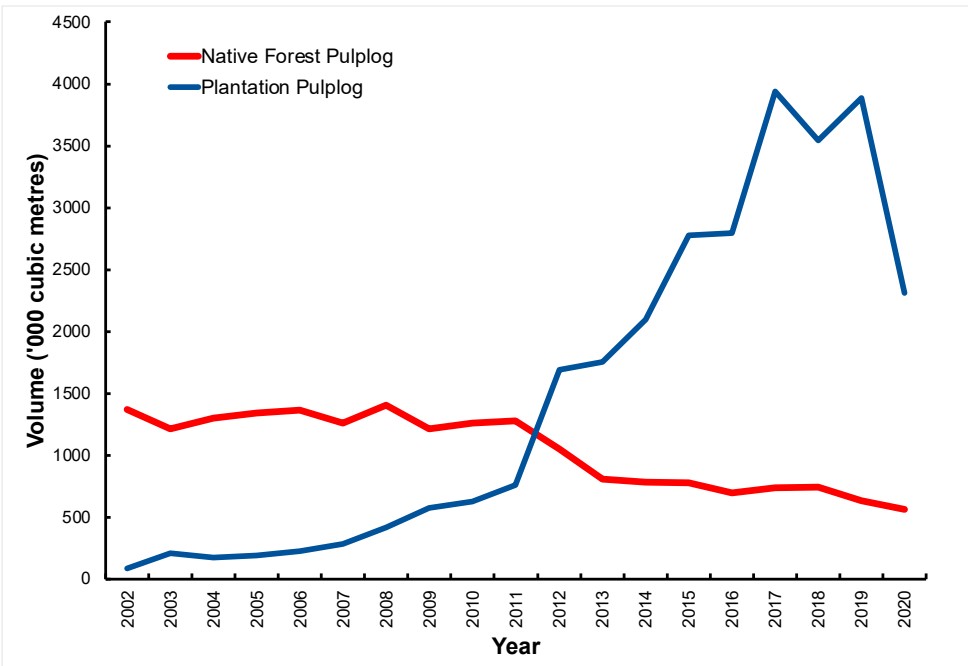

**Figure 4.** Volume of hardwood pulp logs from native forests and plantations (derived from ABARES 2021 data).

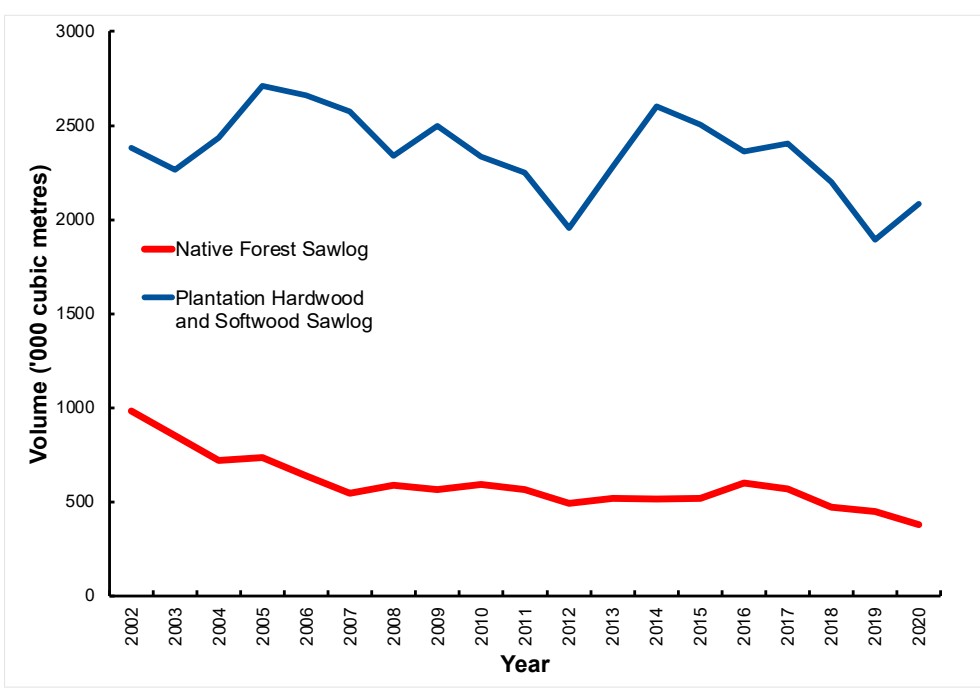

**Figure 5.** Volume of sawlogs from native forests compared with combined volume of sawlogs from softwood and hardwood plantations (derived from ABARES 2021 data).

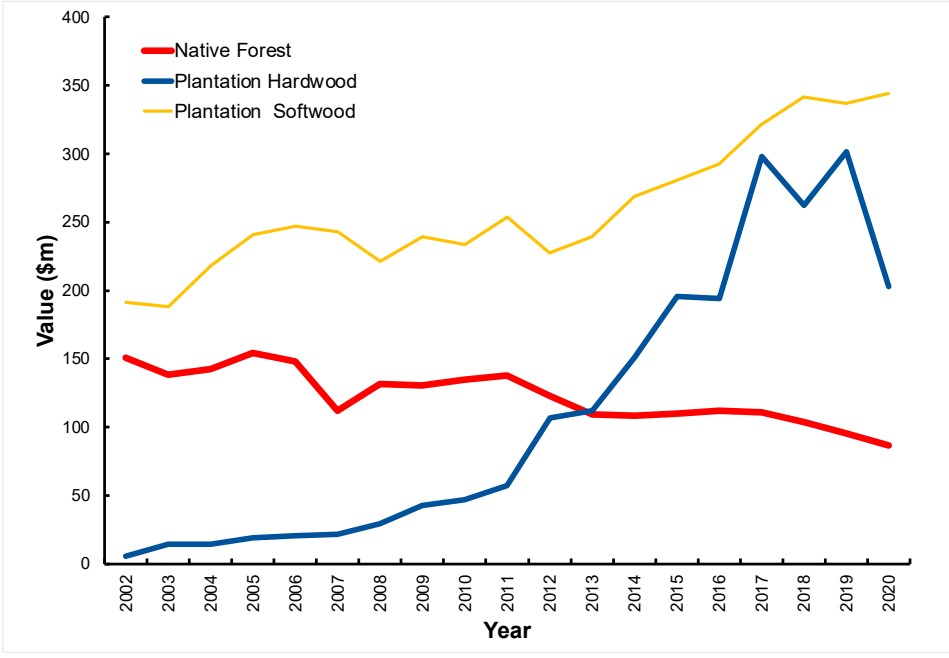

**Figure 6.** Value comparison between logs extracted from native forests, hardwood and softwood plantations (derived from ABARES 2021 data).

The expansion of the Victorian hardwood plantation estate in the past two decades has resulted in significant growth in hardwood pulp log production [96]. Most of this volume is exported as low value unprocessed product [100] that Schirmer, Mylek, Magnusson, Yabsley and Morison [98] suggest is leading to a major loss of timber and pulpwood processing jobs in Victoria. There is therefore considerable potential for the replacement of wood cut from native forests such as those dominated by mountain ash with plantation timber. Most of the wood harvested from Victorian native forests (~86%), including from stands of mountain ash, goes into the pulp and paper stream ([96], including for paper manufacturing. Only

~14% of what is cut from the forest goes into sawn timber production. The potential for substitution of native forest pulp log by plantation hardwood pulp logs has been extensively researched [101–103]. The largest paper manufacturer in Victoria is Australian Paper, and a transition to 100% plantation hardwood for its mills would require around 600,000 m$^3$ of native forest pulp logs. Given the extent of Victoria's hardwood plantation estate and the volume of eucalypt pulpwood logs that are produced, this transition has been deemed technically feasible for more than a decade [103]. The availability of such plantation timber averts the risk of forest industry collapse, as occurred in corporate entities such as the now defunct company Gunns Ltd. (Launceston in Tasmania, Australia) (see [104]). For example, since the majority of hardwood plantation trees to replace the native forest input to Australian Paper's pulp mills in eastern Victoria would be Tasmanian bluegum (*Eucalyptus globulus*), there are increased processing benefits for a transition to hardwood plantations. This is because Tasmanian bluegum has a higher basic density (kg dry fibre/m$^3$) compared with the native forest ash species, which gives the pulp logs more 'dry tons' of weight (kg) per cubic meter of wood [103].

There would be other benefits derived from transitioning from industrial pulp log production from native forests to plantations. For example, greenhouse gas emissions from the land sector can be reduced when wood products are sourced from plantations rather than from native forests [5]. The financial benefits to the Victorian economy of an exit from industrial native forest logging were outlined above [29,85]. Finally, because the primary objective of plantation management is wood production, plantations provide a greater certainty of access to resources than native forests. Moreover, because the rotation time in plantations can be 14–25 years (depending on species, timing and rates of stand entry for thinning and final clear-cut in southeastern Australia), there is a greater chance of extracting a crop of pulp logs in the event of frequent wildfire than in native forests where it can take up to 80+ years to grow sawlogs (see [44]). Notably, landscape-scale simulation of fire occurrence in landscapes with radiata pine (Pinus radiata) plantations in southern New South Wales suggest that plantations may experience much longer intervals between fire in the future, even accounting for future shortening of intervals across all forest types [105].

## 4. Conclusions: General Lessons for Forest Landscape Management

Our case study of mountain ash highlights how following inappropriate policies has led to a loss of options for diversifying forest management. The work has some important general implications for diversifying forest landscape management. First, forest management strategies that are evidently ecologically sustainable must, by definition, account for uncertainty in resource availability such as those caused by disturbances. In the case of mountain ash forests, changes in fire regimes linked with climate change [44], coupled with logging-related increases in forest fire severity [65], will have marked effects on log yields, but these problems have been largely overlooked by policy makers. Indeed, mountain ash forests have been managed with very little to no contingency for wildfire impacts on wood yields. Another area of uncertainty is the impacts of new knowledge about biodiversity on resource availability. For example, long-term data have highlighted the marked temporal declines in populations of species of conservation concern such as Leadbeater's possum and the southern greater glider [47]. This has, in turn, highlighted a need to exempt remaining areas of unlogged and unburned forest from logging to strengthen protection for these taxa, as recognized in a number of court cases successfully prosecuted against the Victorian government logging agency (e.g., [106]).

Second, well informed decisions about diversifying land management require a deep ecological, economic and social understanding of the target ecosystem. This understanding encompasses the condition of the forest ecosystem in question (e.g., the amount of old growth cover relative to historical levels), the status of biodiversity (including species of conservation concern), the integrity of key ecological processes (e.g., fire regimes), levels of timber resource availability and the impacts of logging on other values (e.g., water supply; [7]). This understanding also needs to extend to knowledge of the potential



for interactions between drivers of ecosystem integrity. For example, in mountain ash ecosystems, logging and fire interact, whereby harvested and then regenerated forests are at increased risk of burning at high severity ([65], Figure 2). This, in turn, limits the chance of forests maturing [30], and reduces timber stocks and sawlog supplies [44].

Third, informed management that is ecologically sustainable needs to consider the combined impacts of all disturbance drivers, including those of a natural and human origin, in a given forest ecosystem. In the case of mountain ash forests, the total disturbance burden in the ecosystem needs to be considered, particularly the effects of fire on timber resource availability. The Victoria government's failure to do this and to reduce timber yields following extensive forest losses following major wildfires in 2009 led to the inevitable overcutting of remaining unburned forest. This has both shortened the life of the native forest logging industry and foreclosed options to diversify landscape management strategies that could have maintained other forest values (e.g., the adoption of alternative silvicultural systems to clear-cutting such as the variable retention harvesting system; see [107]). Documents held by the government of Victoria indicate there were concerns about the rate of overcutting in the native forest logging industry as far back as the early 1990s. There also have been long-held concerns about widespread regeneration failure [108]. In fact, the government of Victoria reduced the levels of cut in their sustained yield calculations in all timber regions statewide, except inexplicably, in large forest management units covering the mountain ash forests of the Central Highlands of Victoria. This highlights the importance of heeding early warning signals in levels of resource availability. Otherwise, future options for decision making can subsequently be foreclosed.

Fourth, decisions about forest landscape management need to be underpinned by truly independent assessments of wood resource availability. Such assessments should be made outside of government agencies and by experts and local working groups that do not have a strongly vested interest in resource industries (see [109] for the rationale for doing this). In the case of the mountain ash forests, there is compelling evidence that the Victorian government failed to act as an independent arbiter of the status of wood stocks. Rather, government-based resource management agencies acted as an "arm of industry" and lobbied for long-term commitments of timber that locked in overcutting of the forest [67].

Finally, multifaceted data and perspectives are needed to help guide informed decision making about diversified landscape management. This can be realized through strategies of decentralization becoming a critical component of the regime of forest management [14]. These include Aboriginal, ecological, economic, and social perspectives. In the case of the mountain ash ecosystem and plantations, these perspectives clearly indicate that: (1) There is very limited capacity to continue logging in the mountain ash forest; (2) Ongoing logging will have major negative impacts on biodiversity, water and fire regimes generating further economic losses; (3) There is currently sufficient plantation feedstock available to replace logs from mountain ash forests, particularly in the paper manufacturing sector; (4) The economic value of natural assets such as water, tourism and carbon far exceeds that of industrial logging in mountain ash forests [29]; (5) There are major financial and environmental advantages to making a rapid transition from logging in mountain ash forests to plantation-based wood production [29]. Decentralized strategies for forest management could help Aboriginal communities assume greater control over their respective lands and determine modes of management specific to their respective areas. These can be aligned or contrasted with other needs of the community relevant to particular regions.

**Author Contributions:** Conceptualization, D.L and C.T.; writing—original draft preparation, D.L.; writing—review and editing, C.T.; project administration, D.L.; funding acquisition, D.L. All authors have read and agreed to the published version of the manuscript.

**Funding:** Some of this research was funded by the Victorian Department of Land Water and Planning, grant number 343995.

**Institutional Review Board Statement:** No ethics approvals were required for the research reported in this article.

**Data Availability Statement:** This is a synthesis article, so no intensive data analysis was conducted. Any datasets mentioned in this paper are publicly available as cited in the reference list.

**Acknowledgments:** The authors wish to acknowledge the Koori First Nations People of our study area, the GunaiKurnai, Taungurung and Wurundjeri people, upon whose respective lands this study was conducted. We wish to acknowledge their Elders past and present. We thank Michael Manton and Per Angelstam for suggesting this article be written and for providing valuable input to earlier versions of the manuscript. Comments from two anonymous reviewers helped improve an earlier version of the manuscript. Tabitha Boyer assisted with many editorial aspects of manuscript preparation.

**Conflicts of Interest:** The authors declare no conflict of interest. The funders had no role in the design of the study; in the collection, analyses, or interpretation of data; in the writing of the manuscript, or in the decision to publish the results.

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
