# Peer review of "Diversifying Forest Landscape Management—A Case Study of a Shift from Native Forest Logging to Plantations in Australian Wet Forests"

_land, doi:10.3390/land11030407_

Round 1
Reviewer 1 Report
The article is about the maximum yield logging of Mountain Ash forests which has ultimately led to degradation of the forest ecosystems and destruction of other values. The article couches this story within an ecological sustainable management paradigm with mixed success because a number of other concepts (uncertainty, sharing/sparing, logging regimes) are introduced which cloud this main theme. I've suggested some changes in the article to help with this in particular a suggestion that section 3.4 could be deleted and I have made suggestions to improve the conclusion.
It is plainly obvious that colonial policies based on maximum sustained yield have simply not worked even in the face of better policies such as ecological sustainability and ecosystem services frameworks.
There seems to be a reluctance sometimes to call a spade a spade, perhaps because of the previous flack that the authors have copped in saying what is plainly and scientifically obvious.
This is not to denigrate the piece. I have made some suggestions below and attached the article with additional edits and comments.
The first paragraph, up to line 33, would benefit from mentioning ecosystems services as a concept. I suggest looking at Taye, F. A., M. V. Folkersen, C. M. Fleming, A. Buckwell, B. Mackey, K. C. Diwakar, D. Le, S. Hasan, and C. S. Ange. 2021. The economic values of global forest ecosystem services: A meta-analysis. Ecological Economics 189:107145.
Taye et al provide an added context to the introduction and importantly note that when industrial plantations have claimed greatest economic value of wood production it is because they have not properly considered trade-offs when ecosystem services are collectively valued. This finding would help support the statement on lines 69 and 71.
Line 45 to 47:Why suggest that ecologically sustainable management is even possible under maximum sustained wood production. It makes more sense to say that regimes under maximum sustained yield compromises other values - this would also naturally flow from an introduction that includes a bit more on ecosystem services. The article would then better flow onto the regulated or normal forest concept which has maximum sustained yield at its core.
Line 103. instead say something like there are very few options for diversifying forest management because maximum wood production has degraded or destroyed other values to the point that the entire ecosystem requires restoration or somesuch. I've also made a comment on the manuscript.
A minor point is the need to create more paragraphs and perhaps more subheadings.

Author Response
Thank you for your constructive feedback. Please find attached our response to your comments.

Reviewer 2 Report
This paper uses a case study of the Australian Mountain Ash forest in the state of Victoria to demonstrate how the use of the "regulated forest" concept used by a government agency primarily influenced by the forest industry can lead to the unsustainable use of a natural resource. I found the ecological and economic evidence presented convincing, but the social/governance suggestions less well supported. A few detailed comments by line number follow (a few headings are included just for my tracking purposes).
200 You begin the paragraph with the statement "The Victorian Government’s vision of regulating the Mountain Ash forest has not produced a forest capable of sustaining a yield of wood products in perpetuity." However in this paragraph you only provide citations relating to effects on non-wood services (not clear that any of these significantly affect wood production). Recommend you tie that sentence to the next paragraph and move the non-wood effects discussion after that.
270 For example, the [average?] distance from uncut areas to a disturbance boundary (a road or logged forest) in wood production Mountain Ash forests is just 71 m [32].
3.1 Transition to decentralized forest management
347 Decentralization is believed to bring politics closer to the people, to increase policy effectiveness and to enhance democratic checks and balances at regional and community levels [21].
From my reading of the governance literature, I don't think simple decentralization of authority can be expected to improve the sustainability of natural resources. I would encourage you to address the fact that the original nationalization of some forestlands in the US (and maybe Australia?) was done to counter unsustainable practices of local (and some non-local) actors. And I would suggest you refine your argument for local governance based on the work of Ostrom (1990 and subsequent), who found that resource sustainability correlated with certain governance characteristics, such as 1) strength of bounds (physical and participation), 2) internalization of effects, 3) adaptability through time.
3.4 Other aspects of a rapid transition to a plantation-only forest industry in Victoria
448 One approach to enhance assessments of some of the range of values of plantations is to ensure they are included in environmental accounting [e.g. 29,94].
This could use a bit of clarification.
4. Conclusions: General Lessons for Forest Landscape Management
464 Our case study of Mountain Ash highlights how an over-emphasis on a highly regulated forest model has led [can lead] to a loss of options for diversifying forest management and 465 instead demanded a transition to plantations for wood production.
I don't believe that a transition to plantations was demanded, more that it seems like the most viable option, if supply must be replaced locally.
Author Response

(The authors gave the same response as above.)
